# Longitudinal and Cross-Sectional Relations Between Early Rise Time Discrimination Abilities and Pre-School Pre-Reading Assessments: The Seeds of Literacy Are Sown in Infancy

**DOI:** 10.3390/brainsci15091012

**Published:** 2025-09-19

**Authors:** Marina Kalashnikova, Denis Burnham, Usha Goswami

**Affiliations:** 1Basque Center on Cognition, Brain and Language, 20009 San Sebastian, Spain; 2Ikerbasque, Basque Foundation for Science, 48009 Bilbao, Spain; 3MARCS Institute for Brain, Behaviour and Development, Sydney, NSW 2145, Australia; denis.burnham@westernsydney.edu.au; 4Centre for Neuroscience in Education, University of Cambridge, Cambridge CB2 1TN, UK; ucg10@cam.ac.uk

**Keywords:** amplitude rise time, auditory processing, dyslexia, developmental language disorder, phonological impairment, literacy

## Abstract

Background/Objectives: The Seeds of Literacy project has followed infants at family risk for dyslexia (FR group) and infants not at family risk (NFR group) since the age of 5 months, exploring whether infant measures of auditory sensitivity and phonological skills are related to later reading achievement. Here, we retrospectively assessed relations between infant performance on a rise time discrimination task with new pre-reading behavioural measures administered at 60 months. In addition, we re-classified dyslexia risk at 60 months and again assessed relations to rise time sensitivity. Participants were re-grouped using the pre-reading behavioural measures as either dyslexia risk at 60 months (60mDR) or no dyslexia risk (60mNDR). Methods: FR and NFR children (44 English-learning children) completed assessments of rise time discrimination at 10 and/or 60 months, phonological awareness, phonological memory, rapid automatised naming (RAN), letter knowledge, and language skills (receptive vocabulary and grammatical awareness). Results: Longitudinal analyses showed significant time-lagged correlations between rise time sensitivity at 10 months and both RAN and letter knowledge at 60 months. Rise time sensitivity at 60 months was significantly poorer in those children re-grouped as 60mDR, and rise time sensitivity was significantly related to concurrent phonological awareness, RAN, letter knowledge, and receptive vocabulary, but not to tests of grammatical awareness. Conclusions: The data support the view that children’s rise time sensitivity is significantly related to their pre-reading phonological abilities. These findings are discussed in terms of Temporal Sampling theory.

## 1. Introduction

The longitudinal Seeds of Literacy (SEEDS) project, conducted with infants at family risk for dyslexia who were acquiring Australian English [1,2,3,4], was designed, in part, to test the hypothesis that the perception of amplitude rise time (ART) and the associated percept of metrical rhythm is integral to individual differences in language development from infancy onwards. This hypothesis was drawn from the Temporal Sampling (TS) theory [5], according to which perceptual deficiencies in ART discrimination result in compromised language development (and eventually, compromised reading). ART discrimination is a key sensory factor in the efficiency of neural speech encoding [6,7]; hence, ART discrimination by infants and children is likely to be important regarding individual differences in language development [8]. SEEDS was based on many prior developmental studies showing that ART discrimination is significantly poorer in school-aged children with impairments in linguistic processing who are learning English, French, Dutch, Hungarian, Spanish, Chinese, and Finnish ([5] for review). These cross-language studies focused on children with developmental dyslexia, which is typically characterised by phonological impairments. Interestingly, school-aged children with developmental language disorder or DLD, whose linguistic difficulties span semantics and syntax as well as phonology, also show reduced ART discrimination [9,10,11,12]. Such data are consistent with a core role for ART discrimination in language acquisition.

In the current study, we report the linguistic development of the SEEDS cohort at 60 months of age. We assess relations between ART discrimination and pre-reading skills both longitudinally and concurrently. For this purpose, we relate rise time discrimination previously assessed at 10 months to behavioural pre-reading measures assessed at 60 months, and rise time discrimination at 60 months to the same pre-reading measures at 60 months. We have previously reported that infants assigned to the SEEDS Family Risk for dyslexia (FR) group (by virtue of having a dyslexic parent) already showed significantly impaired ART discrimination compared to No Family Risk (NFR) infants by 10 months of age [1]. Moreover, individual differences in ART sensitivity predicted vocabulary development by 3 years. The FR infants and toddlers have subsequently shown a range of phonological and rhythm-based differences in prior language and other assessments administered between 18 months and 4 years [2,3,4]. To our knowledge, infant measures of sensitivity to ART have neither been related to phonological outcomes in the pre-school period, nor to dyslexia risk status prior to beginning school. This study aimed to fill this gap.

Studies of ART sensitivity in pre-school children are increasingly prevalent in the literature. These have already shown that individual differences in ART discrimination in typically-developing pre-schoolers with no family risk for dyslexia in English and Dutch predict individual differences in phonological skills, both concurrently and longitudinally (aged 3 to 6 years old, [13,14]). ART discrimination is also significantly poorer in pre-school children aged 3 to 5 years at family risk for developmental dyslexia who are learning Dutch [15], a finding corroborated by neurophysiological ART assessments [16]. These and other studies are consistent with TS theory [5], an acoustic theory of individual differences in language development that focuses on developmental relations between ART, phonology, and rhythm. TS theory is based on adult auditory neuroscience data showing that cell networks ‘sample’ acoustic input in parallel in different temporal integration windows and then bind their outputs together to create the human percept that is the speech signal ([17] for review). TS theory proposes that this sampling process (the automatic alignment of endogenous brain rhythms with rhythm patterns in speech) is atypical for children with dyslexia from birth, and perhaps also for children with DLD [18]. Atypical sampling is thought to arise in part because the sensory cues (ARTs) that trigger automatic neural alignment are perceived poorly. Accordingly, the perceptual organisation of speech information is atypical, leading to phonological impairments and reading difficulties. Based on the TS theory, we would expect that the infant measures of ART sensitivity assessed at 10 months should predict individual differences in the pre-reading measures assessed at 60 months. We would also expect developmental continuity in ART sensitivity, with at-risk children who are assessed as having dyslexia risk according to the behavioural assessments at 60 months continuing to show impaired ART discrimination. These specific predictions are tested here.

There are already longitudinal infant FR studies showing that impairments in auditory processing skills predict later language and reading outcomes [19,20,21,22,23,24]. However, no prior infant FR study has focused on ART. Most early longitudinal FR studies focused instead on speech sounds (e.g., bAk/dAk) and/or non-speech tones varying in frequency or duration. These studies reported atypical auditory processing for a wide range of such stimuli from as young as 2 months of age, typically documented using Event Related Potentials (ERPs). These findings of early auditory deficits are supported by the results of more recent studies. For example, Cantiani et al. [25] reported impaired frequency discrimination in Italian infants at FR for either language difficulties or dyslexia. Frequency discrimination was predictive of language scores at 20 months on a parent-estimated vocabulary measure. Mittag et al. [26] measured neural responses to amplitude-modulated white noise (a complex acoustic stimulus sharing rhythmic features with complex speech signals) at six and 12 months of age in a sample at FR for dyslexia, and measured language outcomes from 13 to 30 months. They reported that left-lateralised mean activation in infancy (measured using magnetoencephalography, MEG) in the FR group predicted the production of more complex syntactic structures from 18 months onwards. Moreover, the FR infants did not show increased neural processing efficiency over time, whereas the NFR infants did. Mittag et al. concluded that risk for dyslexia is manifested in early basic auditory processing deficits. The 60-month SEEDS data presented in the current study enable investigation of whether ART, which is important for children’s processing of amplitude-modulated noise [27], is one such basic auditory processing parameter.

The behavioural measures selected for the 60-month assessments reported here were based on an extensive prior literature concerning pre-school predictors of reading development. In order to assess emerging literacy skills in our pre-schoolers, we measured phonological awareness, phonological memory, rapid automatised naming (RAN), and early letter knowledge. A causal association between phonological skills and learning to read has been both hypothesised and documented since the late 1980s [28,29,30]. Wagner and Torgesen noted that three domains of phonological processing contributed to the prediction of later reading: phonological awareness, phonological memory, and RAN. For younger (pre-school) children, phonological awareness tasks with low cognitive load are preferred, such as sound matching and blending [31]. Accordingly, we used a standardised instrument based on such tasks, the Comprehensive Test of Phonological Processing (CTOPP [32]). The CTOPP measures phonological awareness by simple oral blending, elision, and matching tasks, for example, asking children to blend syllables (hot + dog = hotdog; bas + ket = basket) or smaller sounds (b + air = bear; ca + t = cat) to make words [32]. Phonological awareness measured by such tasks in the age range of 2 to 6 years is known to predict individual reading outcomes [31]. The CTOPP also measures phonological memory and RAN in this age range. Early letter knowledge was also assessed because it is one of the most robust predictors of reading development in children [33,34,35,36,37]. Further, studies across languages find early delays in letter knowledge for children who are at-risk for reading difficulties and dyslexia [38,39,40]. Accordingly, infant ART may be expected to predict phonological skills and letter knowledge in pre-reading children.

We also measured vocabulary knowledge at 60 months using standardised tests of receptive vocabulary and composite receptive and productive vocabulary. The literature on vocabulary delays in children with dyslexia is mixed. Prospective FR studies show that receptive vocabulary is affected in 17-month-olds acquiring Dutch [41] and in 3-year-olds acquiring English [42]. For our SEEDS sample, the FR group did not show significantly poorer vocabulary development at 3 years. However, the FR group did show significantly poorer vocabulary than the NFR group at 4 years in one of the two vocabulary measures used for the current assessment (composite receptive and productive vocabulary from the Knowledge Vocabulary sub-test drawn from the Routing–Verbal Domain of the Stanford–Binet Intelligence Scales, [43], see [4]). Previous studies suggest that vocabulary deficits in FR children are most evident in the infant and pre-school stages. By school age, there are typically no longer significant differences in oral vocabulary skills between school-aged children with dyslexia and their typically-developing peers [44]. It is thus possible that the differences in vocabulary skills between FR and NFR children become attenuated by 5 years of age. Given these findings, we also examined whether the vocabulary deficits documented at 4 years of age in the SEEDS sample would persist at 5 years, and whether there would be a relation between children’s oral vocabulary skills and rise time discrimination performance.

In order to explore whether any of the FR sample had co-occurring language difficulties (which would suggest that our SEEDS participants were at risk for DLD rather than pure dyslexia), we also administered two standardised language tasks typically used with DLD samples, the Recalling Sentences sub-test from the Clinical Evaluation of Language Fundamentals (CELF [45]), and the Test of Receptive Oral Grammar (TROG-2 [46]). Children with DLD have difficulties in learning, understanding, and using spoken language, and these tests help to identify whether a child requires a diagnosis of DLD. For example, a DLD child aged 4 years may produce speech more like a two-year-old, using simplified grammar and limited vocabulary [47]. In particular, while dyslexia is identified on the basis of impaired phonology, children with DLD show impairments in semantics and syntax as well as phonology (CATALISE [47]). Both disorders of development are heritable [48]. Our recruitment procedure defined FR for inclusion in SEEDS on the basis of parental dyslexia, and accordingly, we did not expect the FR 60-month-old children to exhibit difficulties in the tasks drawn from the CELF and the TROG. We also did not expect differences in ART discrimination to associate with these tasks, given that our sample did not have a FR for DLD. Finally, our study included a measure of non-verbal IQ. This was not expected to relate to individual differences in acoustic discrimination, but it was controlled in all our analyses involving children’s performance in standardised tests.

### The Present Study

As mentioned above, this study followed the linguistic development of the SEEDS cohort from 10 to 60 months of age. We included measures of ART discrimination at 10 and 60 months, as well as a comprehensive assessment of pre-reading skills at 60 months and a preliminary assessment of dyslexia risk at 60 months.

In this study, the SEEDS sample was classified into FR and NFR groups based on the presence or absence of a family history of dyslexia (specifically, having a dyslexic parent). The administration of the pre-school assessments enabled us to also re-classify the sample at 60m with respect to dyslexia risk by taking into consideration children’s individual pre-reading scores. Thus, we were able to re-classify children as at-risk for a dyslexia diagnosis at 60 months (60mDR) and not at-risk for a dyslexia diagnosis at 60 months (60mNDR), regardless of their dyslexia family risk status at birth.

We then explored infant ART sensitivity and behavioural performance for these two sets of groups—FR vs. NFR and 60mDR vs. 60mNDR. For the FR versus NFR comparisons, we expected that the FR group would show worse performance on the pre-reading behavioural measures than the NFR group. We also expected that the FR group would show poorer ART sensitivity as infants. For the 60mDR versus 60mNDR comparisons, developmental continuity regarding ART sensitivity was predicted. We thus expected that the newly-classified 60mDR would show poorer rise time discrimination as infants compared to the 60mNDR group. Further, we expected that at 60 months, ART sensitivity would be significantly poorer in the 60mDR group than in the 60mNDR group. Finally, we expected significant (40–60%) overlap between the members of the FR group and the 60mDR group.

In line with TS theory, we expected to observe significant concurrent and longitudinal relations between the ART measures and children’s pre-reading skills at 60 months, across the FR and NFR groups. Specifically, we hypothesised that ART discrimination measured in infancy and pre-school age would be related to children’s performance in the letter knowledge, phonological awareness, and RAN tests. On the other hand, we did not predict such associations with the measures of grammar (CELF and TROG) and non-verbal IQ, given that the SEEDS cohort was not at risk for DLD, and that ART discrimination is an automatic sensory process unrelated to general intellectual abilities.

## 2. Materials and Methods

### 2.1. Participants

All children were recruited via the Seeds of Literacy (SEEDS) longitudinal project that investigated language development in children at family risk (FR) and not at family risk (NFR) for dyslexia from 5 months to 5 years of age. To be included in the present study, children were required to (1) have completed the full set of pre-reading behavioural tests at 60 months and (2) have contributed analysable data for at least one measure of rise time discrimination (at 10 months and/or at 60 months). Forty-three of the SEEDS children met these criteria (24 female, 19 male): 20 were originally at risk for dyslexia (FR) and 23 were not at risk (NFR). Two additional children satisfied these inclusion criteria but were excluded from the final sample due to hearing loss (1) and risk or an existing diagnosis of a language disorder other than dyslexia (2). All remaining children were growing up in English-language dominant families, had no hearing impairment, and were not at risk for any other developmental disorders. Children in the FR and NFR groups came from families with comparable socio-economic backgrounds estimated based on the average income level in their area of residence (FR range = levels 10 to 12; median = level 11; NFR range = levels 8 to 12, median = level 11; Mann–Whitney U = 247.5, *p* = 0.661, Rank–Biserial Correlation = 0.076) and maternal education level (FR range = high school degree to PhD, median = university degree; NFR range = high school degree to Master degree; median = university degree; Mann–Whitney U = 209.0, *p* = 0.597, Rank–Biserial Correlation = −0.091).

Participating families were living in Sydney (Australia) and surrounding areas at the time of the study. Caregivers had volunteered to be part of an infant laboratory database, and they were invited to take part in this longitudinal study based on their infants’ age and dyslexia history status. Children’s FR and NFR group assignment at the beginning of the project was based on their parents’ existing dyslexia diagnosis and/or performance on a comprehensive parental screening battery that included language, reading, and cognitive tasks. Both parents of each child completed this battery when the child joined the SEEDS project at 5 months of age. Based on this screening, a child was allocated to the FR group if one of their parents (1) obtained a score of 1.5 *SD* below the average on a measure of word or non-word reading and in at least two of the following tests—oral reading (accuracy, fluency, and rate), spelling, rapid picture naming (RAN), and digit span; (2) indicated history of experiencing reading difficulties in childhood; and (3) obtained an average score (within 0.5 *SD* from the standardised mean) on a measure of non-verbal IQ. A child was allocated to the NFR group if both their parents obtained scores within 0.5 *SD* from the average on all screening tests. Detailed information about parents’ screening performance is provided in the Appendix A.

At the age of 60 months, children completed a set of pre-reading measures that included tests of phonological processing, letter knowledge, grammatical competence, vocabulary, and non-verbal IQ. Each test is described in detail below. The purpose of these measures was two-fold: to explore their relations with previous infant and concurrent child rise time measures, and to screen the children for early indicators of dyslexia at 60 months regardless of their original family risk status. Based on these behavioural tests, a child was now considered to be in the 60-month risk group for dyslexia (60mDR group) if they (1) obtained a score of 1.5 SD below the average on at least two sub-tests of the phonological awareness battery *and* on the letter knowledge test, and (2) obtained an average score (within 0.5 SD from the standardised mean) on all remaining measures. Children who did not satisfy these criteria were now considered to have no risk for dyslexia at 60 months (60mNDR group). Note that these groups were formed solely for the purposes of this study, and they do not reflect children’s future dyslexia status. Full screening for dyslexia or formal diagnosis of dyslexia was not possible, since at the age of 60 months, our participants were about to start formal education and had not yet started learning to read.

As specified in our inclusion criteria above, a child was included in the analyses if they contributed at least one rise time measure (at 10 months or at 60 months). As a result, the sample sizes included in each statistical analysis differed depending on the analysed measures, and accordingly, the data are analysed independently. Data for the rise time task previously administered at 10 months were available for 24 children out of the total 43. Data for the rise time task at 60 months was available for 31 children out of the total 43. Only 12 children contributed data for both rise time measures, so no analyses relating rise time performance at the two time points were conducted (for these 12 children, M thresholds at 10 months were: FR 166.0 msec [SD 40.3], NFR 123.52 msec [SD 47.9]; and at 60 months: FR 36.0 msec [SD 1.0], NFR 23.62 msec [SD 12.2]). All 43 children contributed data for the 60-month pre-reading measures. The specific breakdown of the sample sizes by measure and group can be found in the Appendix A.

### 2.2. Amplitude Rise Time (ART) Discrimination Tasks

#### 2.2.1. Infant Task (10 Months)

This was an infant visual preference version of a two-alternative forced-choice (2AFC) adaptive threshold procedure [49] designed specifically for the SEEDS project [1].

During this task, infants sit facing three computer monitors. The central monitor is used to present a visual attention getter between trials, and the left and right monitors are used to present images of colourful checkerboard patterns during the experimental trials to measure infants’ visual attention in response to the auditory stimuli associated with each side. The auditory stimuli consist of strings of sinewave tones (500 Hz) with different rise times (ranging from 15 ms to 300 ms). One string is the repeating string, and it consists of a repeating tone with the same rise time (15 ms, 15 ms, 15 ms, 15 ms, etc.). The other is the alternating string, and it consists of alternations of the tone from the repeating string and another tone with a longer rise time (e.g., 15 ms, 300 ms, 15 ms, 300 ms, etc.). For each infant, each string is assigned to one of the sides of the display (left or right) for the duration of the task (a total of 25 trials). At the start of each trial, infants are presented with the visual stimuli in silence. When they direct their gaze to one of the sides, they hear the string that was assigned to that side. Critically, this task is adaptive, so the maximum rise time used for the alternating string is manipulated according to the infants’ performance. Greater (≥55%) fixation to the alternating stimulus side for two consecutive trials resulted in a step down (e.g., 15 ms, 270 ms, 15 ms, 270 ms, …), and less than 55% to a step back up (i.e., a reversal occurred every time that the steps changed direction) on the alternating side sound but no change to the repeating side sound.

The adaptive nature of this task allowed us to calculate individual thresholds for rise time discrimination calculated as the difference between the two rise times in the alternating string presented for the last three step reversals. Thus, higher thresholds indicated poorer rise time discrimination. Please see Kalashnikova et al. [1] for further details on the apparatus, technical parameters of this task, and procedure for data pre-processing.

#### 2.2.2. Child Task (60 Months)

This was an adaptive staircase AXB rise time discrimination task commonly used with pre-school and school-aged children [13,50,51]. Forty pure sine tones were used as auditory stimuli. The tones were 800 msec in duration and were matched for fundamental frequency (500 Hz), and fall time (50 m s), but differed in onset rise times, which ranged from 15 msec to 300 msec in 15 msec intervals. The task was conducted in a quiet child-friendly room inside an infant laboratory. The child was seated on a chair facing a computer monitor. An experimenter sat next to the child and controlled the computer mouse to register the child’s responses and advance the trials. Auditory stimuli were presented over loudspeakers located on the left and right sides of the monitor.

The “Dinosaur game” threshold estimation program created by Dorothy Bishop (Oxford University, 2001) was used to administer the task. In this task, the child sees three cartoon dinosaurs and is told that the dinosaurs will make different sounds. The experimenter explained that two dinosaurs would make the same sound (standard stimulus), and one would make a different sound (alternating stimulus), and it was the child’s task to find the dinosaur who made the different sound (in this case, the dinosaur associated with the slowest rise time). The specific stimuli presented on each trial were determined by the child’s performance. This task uses the more virulent form of Parameter Settings by Sequential Estimation (PEST [52]) to adapt the steps based on the child’s answer on a previous trial. Children were also provided with feedback on each trial: children collected a sticker for every correct response and an image of a red ‘X’ for every incorrect response. All children completed 40 trials. The dependent variable was the rise time discrimination threshold computed as the average rise time of the alternating stimulus for the last four reversals for a child.

### 2.3. Child Pre-Reading Behavioural Tests (60 Months)

#### 2.3.1. Phonological Processing

Children completed the Comprehensive Test of Phonological Processing (CTOPP) [32]. This is a standardised test of phonological abilities. It includes four sub-tests that separately assess phonological awareness (tasks of elision, blending, and sound matching), phonological memory (tasks of memory for digits and non-word repetition), symbolic RAN (rapid letter and digit naming), and non-symbolic RAN (rapid colour and object naming). A standardised score (M = 100; SD = 15) was computed for each sub-test.

#### 2.3.2. Sentence Repetition

Children completed the Recalling Sentences sub-test from the Clinical Evaluation of Language Fundamentals (CELF) [45]. This is a standardised test of grammatical competence. In this test, children hear a sentence and are required to repeat it verbatim. Responses are scored according to the number of errors made in each repetition and used to compute a standardised score (M = 10, SD = 3).

#### 2.3.3. Knowledge of Grammar (TROG)

Children completed the Test of Reception of Grammar (TROG-2) [46]. This is a standardised test of children’s understanding of different grammatical constructs. In this test, children see a page with four images, hear a sentence, and are required to point to the picture corresponding to the sentence. A standardised score is computed (M = 100; SD = 15) for the entire test.

#### 2.3.4. Letter Knowledge

Children completed the letter knowledge section from the early reading skills subtest of the Wechsler Individual Achievement Test, WIAT III [53]. In this test, children saw a booklet depicting letters from the alphabet and were asked to name some letters and find some letters named by the experimenter. Each child completed 13 test items and received a score of 1 for a correct response on each item, so the total scores could range from 0 to 13.

#### 2.3.5. Receptive Vocabulary

Children completed the Peabody Picture Vocabulary Test (PPVT) [54]. This is a standardised test of receptive vocabulary. In this test, a child sees a page with four images, hears a word, and is required to point to the picture corresponding to the word. A standardised score is computed (M = 100; SD = 15) for the entire test.

#### 2.3.6. Composite Vocabulary

Children completed the Knowledge Vocabulary sub-test in the Routing–Verbal Domain of the Stanford–Binet Intelligence Scales-5th Edition [43]. This test assesses children’s receptive and expressive vocabulary skills. In the receptive vocabulary items, children hear a word and are asked to point to its referent on a picture (or on themselves in case of body parts), and in the expressive vocabulary items, children hear a word and see its image and are asked to provide the word’s definition. A single scaled score (M = 10; SD = 3) is computed for this sub-test.

#### 2.3.7. Non-Verbal IQ

Children completed the Fluid Reasoning Object Series/Matrices sub-test in the Routing Non-Verbal Domain of the Stanford–Binet Intelligence Scales-5th Edition [43]. A single scaled score (M = 10; SD = 3) is computed for this sub-test.

## 3. Results

### 3.1. Child Pre-Reading Behavioural Tests at 60 Months

Table 1 presents the FR (N = 20) and NFR (N = 23) children’s scores on the behavioural tests (full sample N = 43) according to their original family risk designation, as well as the results of independent-samples *t*-tests comparing group performance. As can be seen, at 60 months of age, NFR children significantly outperformed the FR children on tests of rapid symbolic naming and letter knowledge. There were no significant group differences regarding performance in the remaining pre-reading behavioural tests; however, there were medium effect sizes in the expected direction for phonological awareness (d = 0.482) and phonological memory (d = 0.599).

### 3.2. Infant Rise Time Discrimination and Pre-Reading Behavioural Tests at 60 Months

A correlational analysis was conducted to assess the retrospective relations between children’s performance in the pre-reading behavioural tests at 60 months and their rise time discrimination thresholds assessed at 10 months of age. Data from 24 children were available for this analysis. Based on the TS theory, we tested the specific prediction that infant rise time thresholds would be negatively correlated with children’s phonological processing, letter knowledge, and vocabulary scores at 60 months (as higher [poorer] rise time thresholds should be related to lower behavioural scores). Thus, one-tailed Pearson correlations were conducted. As can be seen in Table 2, infant rise time thresholds were significantly negatively correlated with scores on the letter knowledge and one of the two RAN tests, failing to reach significance for the second RAN test (non-symbolic RAN task, *p* = 0.011, symbolic RAN task *p* = 0.051). The significant correlations are depicted in scatterplots in Figure 1.

### 3.3. Infant Rise Time Discrimination at 10 Months and 60-Month Dyslexia Risk Status: A Retrospective Analysis

The developmental continuity of impaired rise time discrimination in the at-risk children was explored retrospectively. To compare performance between the two dyslexia risk groups based on the 60-month pre-reading behavioural tests (60mDR vs. 60mNDR groups), an independent-samples *t*-test was conducted (Figure 2), comparing the 10-month-old rise time thresholds between children subsequently placed in the 60mDR group and children placed in the 60mNDR group (total N = 24). Given the large differences in sample sizes and variances between groups, a Welch *t*-test was used, and it revealed no significant group differences, t(4.613) = −0.971, *p* = 0.379, d = −0.514 (see Figure 2).

### 3.4. Developmental Continuity: Child Rise Time Discrimination at 60 Months

Preliminary screening of the data revealed that 60-month rise time discrimination thresholds were negatively skewed, so they were not suitable for parametric statistical analyses. Therefore, the raw rise time thresholds were rank normalised, which resulted in a normal distribution of scores (see Appendix A for more details about the rank transformation).

Two sets of independent-samples *t*-test analyses were conducted: the first compared performance at 60 months between the two family risk groups as designated at the outset of the SEEDS project (FR vs. NFR, see Figure 3A), and the second compared performance between the two dyslexia risk groups based on their re-classification following the 60-month pre-reading behavioural measures (60mDR vs. 60mNDR, see Figure 3B). Theoretically, the FR and 60mDR children should show higher (worse) rise time thresholds, accordingly one-tailed tests were applied. Numerically, FR children indeed obtained higher mean rise time discrimination thresholds than NFR children when assessed at 60 months; however, this difference just missed statistical significance, t(29) = 1.677, *p* = 0.052, d = 0.610, although it showed a medium effect size. Similarly, when the thresholds were compared between children placed in the dyslexia risk group at 60 months and children placed in the no dyslexia risk group at 60 months, 60mDR children obtained numerically higher rise time discrimination thresholds compared to the 60mNDR children, a medium-sized effect that did reach statistical significance, t(15.489) = 1.830, *p* = 0.044 d = 0.609 (Welch *t*-test was used given the large difference in sample sizes and variances).

### 3.5. Child Rise Time Discrimination and Pre-Reading Behavioural Tests at 60 Months

Finally, a cross-sectional correlational analysis was conducted to assess the concurrent relations between children’s performance in the pre-reading behavioural tests and their rise time discrimination thresholds at 60 months. Group performance in the pre-reading tasks at the 60-month re-grouping for those children who also contributed rise time thresholds at 60 months is shown in Table 3 (please note, 31 children contributed data for this analysis, in contrast to the 43 children in Table 1; see also SM2). The correlations are shown in Table 4. Comparison of Table 3 with Table 1 shows that phonological awareness and phonological memory skills were significantly poorer in this more focused sample, and that language scores (CELF, TROG, receptive vocabulary) and non-verbal IQ were also poorer. Further inspection of the data revealed that none of the individual 60mDR children had risk scores on the TROG, CELF, PPVT, and Stanford–Binet (risk score = 1.5SD below the mean); however, as a group, they had lower scores than the 60mNDR children. This suggests that while the 60mDR children do not qualify as DLD + dyslexia, the 60mDR children do have overall lower language abilities than children who have no family risk for dyslexia or children who have strong phonological skills.

As we were testing for the negative correlations predicted by Temporal Sampling theory, one-tailed Pearson correlations were conducted. Regarding the cross-sectional correlations, it was expected that rise time thresholds would be negatively correlated with children’s phonological processing, letter knowledge, and vocabulary scores, but not necessarily with their TROG or CELF scores. As can be seen in Table 4, rise time discrimination thresholds were indeed significantly negatively correlated with scores on the phonological awareness, symbolic RAN, non-symbolic RAN, and letter knowledge measures. The significant relations are shown as scatterplots in Figure 4. The scatterplots show that one 60mDR child did not follow the associations shown at the group level (see circled data point). This child’s profile is also discussed further in the Discussion.

## 4. Discussion

This work addressed the overarching question of whether the seeds of later individual differences in literacy are sown in infancy. For this purpose, we assessed sensitivity to ART in infancy (at 10 months) and in pre-school (at 60 months) in children at and not at family risk for dyslexia, as well as their performance on assessments of pre-reading abilities in pre-school. Our predictions were based on the TS theory. We expected ART sensitivity to show developmental sensitivity, so it was predicted to be poorer in the FR than the NFR group at 10 months and in the 60mDR than the 60mNDR group at 60 months. We also hypothesised time-lagged and concurrent relations between ART sensitivity measures and children’s performance in pre-reading measures (spanning phonological awareness, RAN, and phonological memory), but not in measures of other language abilities (related to syntax and grammar) and non-verbal IQ. The findings related to each of these predictions will now be discussed in more detail.

Regarding the longitudinal predictions based on the TS theory, infant sensitivity to ART at 10 months was found to be significantly related to letter knowledge at 60 months and to RAN (moderate-size correlations; see Table 2). Although the other phonological measures (phonological awareness and phonological memory) did not show significant longitudinal relations with this infant ART measure, it is quite remarkable that individual differences in letter knowledge and in non-symbolic RAN were predicted by auditory sensitivity to ART measured over 4 years earlier. Non-symbolic RAN tasks involve picture and colour naming, and the FR group was slower than the NFR group at producing these overlearned phonological forms at 60 months (effect size 0.456, Table 1). These relations between ART sensitivity and phonological skills as measured by RAN and letter knowledge are supportive of a developmental trajectory relating early auditory sensitivity to phonological development that is also documented by our prior SEEDS findings [1,2,3,4].

Regarding the expectation of developmental continuity of ART sensitivity between infancy and 60 months, this was broadly supported by comparison of the group means (see Figure 1 and Figure 3). Unfortunately, participant attrition over the 5 years meant that sample sizes were rather small, a limitation to which we return below. The retrospective re-grouping of the 10-month-old data on the basis of dyslexia risk at 60 months did not reveal significantly lower ART thresholds in the 60mNDR group compared to the 60mDR group (Figure 2). However, when re-assessed for dyslexia risk and ART sensitivity at 60m, the two risk groups did differ in ART sensitivity (Figure 3B). Indeed, at 60m, the original FR group showed a trend to have poorer ART sensitivity than the NFR group (effect size 0.610). Accordingly, there is partial support for developmental continuity.

Meanwhile, it was clear from the scatterplots provided as Figure 1 and Figure 4 that occasionally infants with high-risk status exhibited low (= good) ART thresholds despite their family risk status for dyslexia. One FR infant showed the second most sensitive ART threshold of all the infants tested at 10 months (see individual data points on Figure 1), despite having a mother with dyslexia. This participant went on to have very high scores for non-verbal IQ, vocabulary, and grammar (TROG test). The child also had phonological awareness skills in the normal range, and after the 60-month assessment, changed status to 60mNDR. The child circled on Figure 4, meanwhile, was originally in the FR risk group because their father had a diagnosis of dyslexia, and this child exhibited a high (poor) ART threshold at 10 months. The child subsequently had poor ART skills at 60 months, yet had good letter knowledge, phonological awareness, and vocabulary skills. The only phonological measure showing poor performance at 60 months was the RAN task. When risk status was re-assigned at 60 months based on the pre-literacy task battery, this child no longer showed a dyslexia risk, moving to the 60mDR group (as shown in Figure 4). For the sample as a whole, when risk status was re-assigned at 60 months, ART sensitivity at 60 months was significantly worse for the 60mDR children compared to the 60mNDR children (Figure 3B). Accordingly, with occasional exceptions, ART sensitivity does seem to show developmental continuity in terms of dyslexia risk status.

Consistent with the developmental patterns proposed by TS theory, the cross-sectional analyses at 60 months showed that individual differences in ART sensitivity at 60 months were significantly related to all the theoretically-driven pre-literacy measures (Table 4, moderate-size correlations): phonological awareness, both symbolic and non-symbolic RAN, and letter knowledge. There was also a trend for sensitivity to ART at 60 months to be related to vocabulary skills at 60 months (small-sized correlations < 0.300). This suggests that ART sensitivity may still affect vocabulary development at the age of 5 years, even though overall vocabulary development was not different between the risk groups by this time point (Table 1). Of the three key domains of phonological processing known from prior work [29] to be predictive of individual differences in literacy (phonological awareness, RAN, and phonological memory), only phonological memory was not significantly related to ART sensitivity. This is not particularly surprising, as many subsequent studies investigating the predictive strength of the three phonological domains originally described by Wagner & Torgesen have found more consistency regarding RAN and phonological awareness compared to phonological memory (e.g., [55]). Sensitivity to ART at 60 months was not related to the two linguistic measures of grammar and syntax that were included in the task battery, the TROG and the CELF (Table 2). Again, this is unsurprising, as the SEEDS sample was not selected for family risk for DLD. As the selected linguistic tasks at 60 months share a range of cognitive and attentional processing mechanisms, the specific relations between ART and tasks that rely on the phonological lexicon are notable.

### Limitations

As is often the case in longitudinal studies with infants and young children, our sample sizes differed by measure and time point and were overall conservative. As a result, some of our statistical analyses may have been underpowered, failing to reach statistical significance. The observed effect sizes provide ground for our interpretations, but we acknowledge that their robustness must be tested in larger samples in the future. This limitation is particularly relevant for our retrospective longitudinal analyses. Only 24 infants provided data for the analyses involving infant ART thresholds at 10 months and performance on the phonological assessments at 60 months. Behavioural infant tasks are associated with large rates of data loss and exclusion of unanalysable data (~30% exclusion rate in this cohort; see Kalashnikova et al. [2,3] for a detailed description of the criteria for inclusion in data analyses). Thus, while this type of behavioural task provides a rapid and relatively inexpensive way for assessing infant auditory abilities, other measures may prove to be more suitable for capturing individual discrimination indices for longitudinal follow-ups. Future studies, for example using neurophysiological methods such as electroencephalography (EEG), may be able to address this limitation and also include a wider range of acoustic parameters in infancy.

A second limitation is that many of the participants who were assessed as FR at the beginning of the study no longer met our criteria for dyslexia risk by the age of 60 months. Only 3 of the 13 FR children who had 60-month data for the rise time discrimination task (23%), and 6 out of 20 who had 60-month data for the pre-reading measures (30%) were classified as risk, which is lower than typical family risk estimates (40–60%). However, it may be that via their participation in the study, parents of FR children became more focused on the factors that create a supportive pre-literacy environment, and hence provided a more optimal pre-school pre-literacy environment than is typical in the home of FR children, reducing their child’s risk.

A final limitation of this study is that until the children are older, it is not possible to be certain that the 60mDR group will actually meet the criteria for a diagnosis of dyslexia. Hence, future re-grouping, taking into consideration children’s actual early reading skills, may lead to different results.

## 5. Conclusions

The relations between sensitivity to ART and phonological development found previously in studies of pre-reading children and older children with developmental dyslexia (e.g., [13,14,15,16,49,56]) were also found in the SEEDS cohort, extending these developmental relations into infancy. In particular, infant ART sensitivity was a significant predictor of both letter knowledge and RAN at 5 years of age. The behavioural assessments administered at 60 months of age also enabled re-assignment of dyslexia risk status, and ART sensitivity at 60 months was found to be significantly poorer in the 60mDR children. These children also showed significantly poorer phonological and language skills compared to 60mNDR children, with phonological awareness, RAN, letter knowledge, vocabulary, and grammar skills at 60 months all showing group differences. However, only phonological awareness, RAN, and letter knowledge showed significant relations with ART sensitivity. Grammatical skills were, however, unrelated to ART sensitivity in these children. Accordingly, the SEEDS cohort can be considered at risk for reading impairments rather than generalised language learning impairments. Infant risk studies that group babies into a single language learning impaired group may thus miss important differences in sensory/neural processing that could help to distinguish later developmental trajectories.

## Figures and Tables

**Figure 1 brainsci-15-01012-f001:**
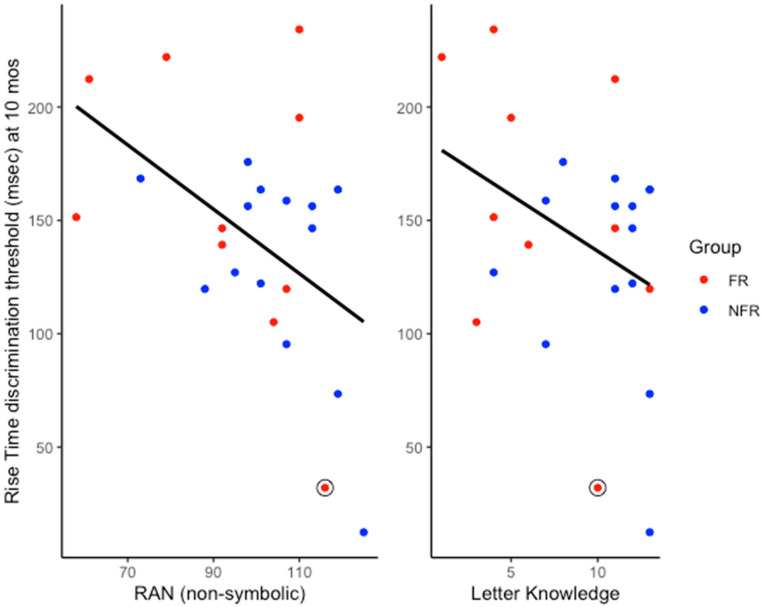
Relations between infant rise time discrimination thresholds at 10 months and non-symbolic ran and letter knowledge scores at 60 months (N = 24). Colours indicate children’s family risk status. See Discussion for the cognitive profile of the FR child who contributed the data point enclosed in a circle.

**Figure 2 brainsci-15-01012-f002:**
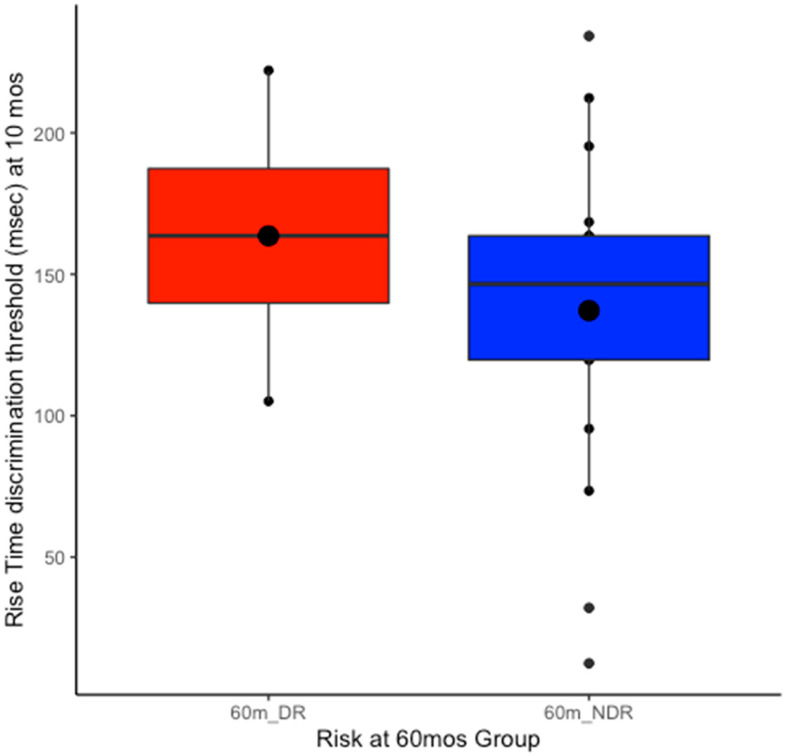
Retrospective exploration of mean rise time discrimination thresholds obtained at 10 months by infants who were subsequently placed in the 60mDR group and the 60mNDR group on the basis of the pre-reading behavioural tests (total N = 24)**.** The figure presents the data after the 60-month re-grouping (disregarding infants’ original group allocation based on family risk). The internal large circles represent the mean, the internal line represents the median, and the hinges extend to the first and third quartiles.

**Figure 3 brainsci-15-01012-f003:**
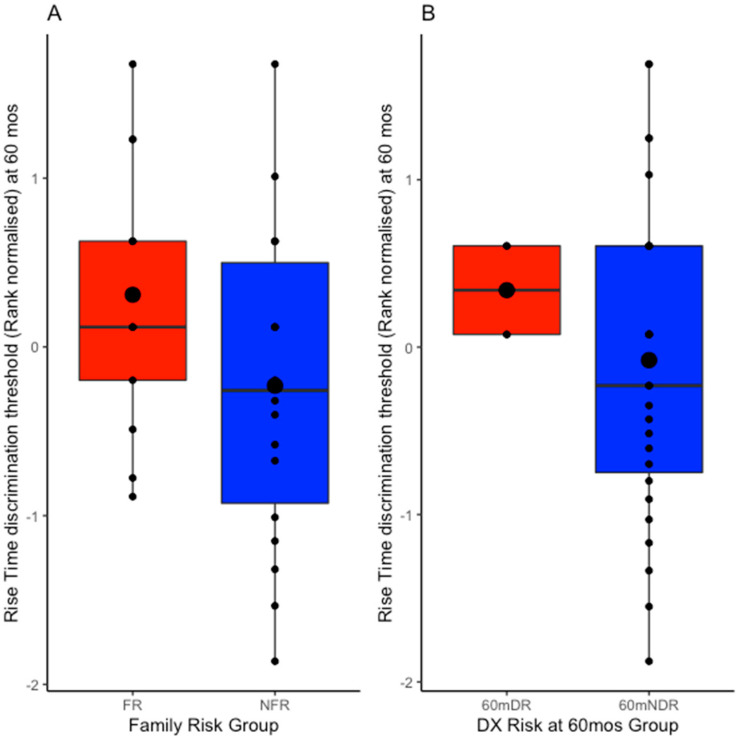
Mean rise time discrimination thresholds obtained when children were 60 months old (total N = 31). Panel (**A**) presents the data for the two family risk groups as designated at the outset of the SEEDS project (FR shown in red and NFR shown in blue). Panel (**B**) presents the data for the two dyslexia risk groups based on their performance on the 60-month pre-reading behavioural measures (60mDR shown in red and 60mNDR shown in blue). The internal large circles represent the mean, the internal line represents the median, and the hinges extend to the first and third quartiles.

**Figure 4 brainsci-15-01012-f004:**
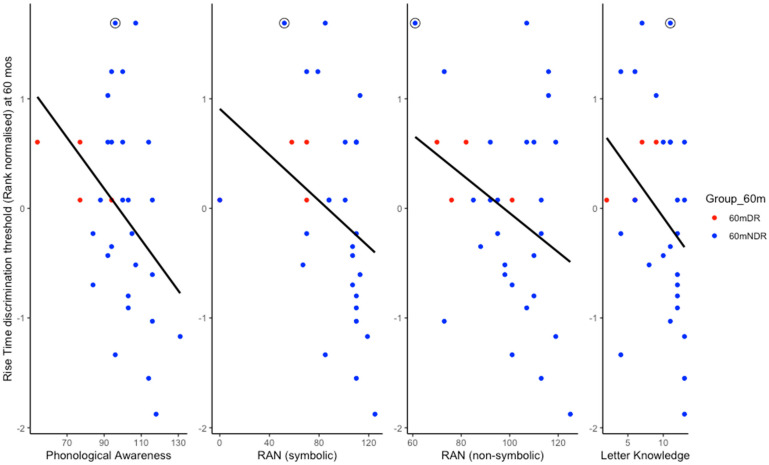
Relations between 60-month rank normalised rise time discrimination thresholds, standard scores for phonological awareness, symbolic RAN, and non-symbolic RAN, and the absolute letter knowledge scores (N = 31). Colours indicate children’s 60m risk status. See Discussion for the cognitive profile of the FR child who moved to 60mDNR and contributed the data point enclosed in a circle.

**Table 1 brainsci-15-01012-t001:** At family risk (FR) and not at family risk (NFR) children’s scores (M (Range; SD)) on the pre-reading behavioural tests completed at 60 months.

	FR	NFR	T ^a^	*p*	Cohen’s d
Phonological awareness	93.9 (54–114; 12.908)	100.7 (71–131; 15.062)	−1.576	0.123	−0.482
Phonological memory	96.15 (70–128; 17.11)	105.826 (85–128; 15.299)	−1.958	0.057	−0.599
**Symbolic RAN (digit and letter)**	**65.6 (0–116; 43.023)**	**98.696 (58–125; 18.154)**	**−3.365**	**0.002**	**−1.029**
Non-symbolic RAN (colour and object)	95.75 (58–116; 18.241)	103.174 (70–125; 14.374)	−1.491	0.143	−0.456
Repeating Sentences	10.85 (6–15; 2.601)	12.043 (7–17; 2.602)	−1.500	0.141	−0.459
**Letter Knowledge**	**7.45 (1–13; 3.79)**	**10.130 (4–13; 3.005)**	**−2.585**	**0.013**	**−0.790**
Reception of grammar	104.75 (69–130; 19.051)	111.043 (88–141; 13.357)	−1.267	0.212	−0.387
Receptive vocabulary	115.1 (85–130; 10.686)	119.652 (97–140; 12.773)	−1.256	0.216	−0.384
Composite vocabulary	10.3 (7–15; 1.895)	10.478 (6–18; 2.826)	0.239	0.812	0.073
Non-verbal IQ	11.8 (5–19; 3.722)	12.304 (7–19; 3.866)	−0.434	0.666	−0.133

Note. Statistically significant results are marked in bold. ^a^ Results of independent-samples *t*-test analyses comparing group performance (N = 43).

**Table 2 brainsci-15-01012-t002:** Pearson correlations between infant rise time discrimination thresholds and scores on the pre-reading behavioural tests at 60 months (N = 24).

Pre-Reading Behavioural Tests at 60 Months	Rise Time Discrimination Threshold at 10 Months	
	**r**	** *p* **
Phonological awareness	−0.248	0.122
Phonological memory	0.079	0.644
Symbolic RAN (letters and digits)	−0.341	0.051
**Non-symbolic RAN (colours and objects)**	**−0.467**	**0.011**
Repeating sentence	−0.132	0.269
**Letter knowledge**	**−0.356**	**0.044**
Reception of Grammar	−0.174	0.208
Receptive vocabulary	−0.172	0.211
Composite vocabulary	−0.235	0.135
Non-verbal IQ	0.020	0.536

Note. Statistically significant results are marked in bold.

**Table 3 brainsci-15-01012-t003:** At-risk (60m_DR) and not at-risk (60m_NDR) children’s scores (M (Range; SD)) on the pre-reading behavioural tests completed at 60 months (N = 31).

	60m_NDR	60m_DR	T ^a^	*p*	Cohen’s d
**Phonological awareness**	**102.185 (84–131; 11.526)**	**75.5 (54–94; 16.422)**	**4.108**	**<0.001**	**2.201**
**Phonological memory**	**107.185 (85–128; 15.237)**	**80.75 (70–92; 9.708)**	**3.343**	**0.002**	**1.791**
**Symbolic RAN (digit and letter)**	**94.704 (0–125; 26.229)**	**49.5 (0–70; 33.481)**	**3.117**	**0.004**	**1.670**
**Non-symbolic RAN (colour and object)**	**101.37 (61–125; 15.564)**	**82.250 (70–101; 13.426)**	**2.324**	**0.027**	**1.245**
Repeating Sentences	12.074 (7–17; 2.63)	9.25 (7–11; 2.062)	2.045	0.050	1.096
**Letter Knowledge**	**9.852 (4–13; 3.06)**	**6 (2–9; 2.944)**	**2.359**	**0.025**	**1.264**
**Reception of grammar**	**110.519 (69–141; 15.851)**	**88 (69–111; 18.221)**	**2.609**	**0.014**	**1.398**
**Receptive vocabulary**	**120 (100–140; 9.77)**	**104.5 (85–116; 13.478)**	**2.832**	**0.008**	**1.517**
Composite vocabulary	10.963 (6–18; 2.752)	9 (7–11; 1.826)	1.372	0.181	0.735
Non-verbal IQ	12.407 (7–19; 3.993)	12.250 (9–16; 2.986)	0.075	0.940	0.040

Note. Statistically significant results are marked in bold. ^a^ Results of independent-samples *t*-test analyses comparing group performance (N = 31).

**Table 4 brainsci-15-01012-t004:** Pearson correlations between rise time discrimination thresholds at 60 months and scores on the pre-reading behavioural tests at 60 months (N = 31).

Pre-Reading Behavioural Tests at 60 Months	Rise Time Discrimination Threshold at 60 Months	
	**r**	** *p* **
**Phonological awareness**	**−0.385**	**0.016**
Phonological memory	−0.121	0.259
**Symbolic RAN (letters and digits)**	**−0.358**	**0.024**
**Non-symbolic RAN (colours and objects)**	**−0.323**	**0.038**
Repeating sentence	−0.192	0.150
**Letter knowledge**	**−0.325**	**0.037**
Reception of grammar	−0.164	0.188
Receptive vocabulary	−0.286	0.060
Composite vocabulary	−0.271	0.070
Non-verbal IQ	−0.261	0.078

Note. Statistically significant results are marked in bold.

## Data Availability

Anonymised and tabulated data can be made available upon reasonable request to the first author due to ethical issues.

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
