# Peer review of "Longitudinal and Cross-Sectional Relations Between Early Rise Time Discrimination Abilities and Pre-School Pre-Reading Assessments: The Seeds of Literacy Are Sown in Infancy"

_brainsci, 2025, doi:10.3390/brainsci15091012_

Round 1

Reviewer 1 Report

Comments and Suggestions for Authors

The article presents a compelling longitudinal investigation into the predictive relationship between infant amplitude rise time (ART) sensitivity and pre-reading abilities at 60 months, framed within the Temporal Sampling (TS) theory. While the study is grounded in a strong theoretical framework and addresses a meaningful gap in the literature by linking early auditory processing to later literacy outcomes, several limitations constrain its overall impact. The introduction, though conceptually thorough, suffers from unnecessary repetition of hypotheses and theoretical expectations, which detracts from clarity and focus. Methodologically, the use of established behavioural measures and a rare longitudinal design beginning in infancy are notable strengths. However, the small sample size, particularly for the key longitudinal analyses (N = 24), is not sufficiently justified, and the absence of important covariates—such as socioeconomic status or home literacy environment—limits the interpretive scope of the findings. Although the dual grouping strategy based on both family risk (FR) and behavioural risk at 60 months (60mDR) adds conceptual depth, its theoretical implications are underdeveloped, particularly in light of divergences between classifications that remain insufficiently explored. The results and discussion sections build coherently on the study’s predictions, confirming associations between ART sensitivity and pre-literacy skills like rapid automatized naming (RAN) and letter knowledge. These findings are consistent with TS theory and suggest ART may serve as an early marker of reading risk. Nevertheless, the interpretation of these associations is at times overstated. The authors frequently imply developmental causality based on correlational data, with limited reporting of statistical details such as effect sizes, model specifications, or confidence intervals. Moreover, findings that barely reach statistical significance (e.g., p = .044, p = .051) are discussed with unwarranted confidence. The discussion rightly acknowledges ART’s specific relation to phonological rather than grammatical skills, supported by the null associations with TROG and CELF measures; however, this specificity is not critically examined in light of alternative explanations, such as shared attentional or cognitive processing mechanisms. While outlier cases are used to illustrate variability in developmental outcomes, they are presented narratively without systematic analysis or integration into the theoretical framework. The discussion would benefit from deeper engagement with competing models, particularly the multiple deficit model, which could offer a more nuanced interpretation of the data than TS theory alone. Critically, the article does not provide a dedicated limitations section, and the few methodological caveats—such as attrition or the limited range of acoustic sensitivity measures—are mentioned only briefly and treated as peripheral rather than central issues. This absence undermines transparency and weakens the reader’s ability to assess the generalizability of the findings. The lack of consideration for environmental or demographic confounds further narrows the explanatory power of the conclusions. Additionally, the prose style, while technically competent, is overly dense and prone to long, complex sentences that may impede comprehension. Overall, the study offers valuable preliminary evidence that ART sensitivity in infancy may relate to later phonological processing and literacy-related outcomes, yet its impact would be significantly enhanced by greater methodological transparency, theoretical pluralism, and a more cautious and critically engaged interpretation of the results.

Comments on the Quality of English Language

The article presents a compelling longitudinal investigation into the predictive relationship between infant amplitude rise time (ART) sensitivity and pre-reading abilities at 60 months, framed within the Temporal Sampling (TS) theory. While the study is grounded in a strong theoretical framework and addresses a meaningful gap in the literature by linking early auditory processing to later literacy outcomes, several limitations constrain its overall impact. The introduction, though conceptually thorough, suffers from unnecessary repetition of hypotheses and theoretical expectations, which detracts from clarity and focus. Methodologically, the use of established behavioural measures and a rare longitudinal design beginning in infancy are notable strengths. However, the small sample size, particularly for the key longitudinal analyses (N = 24), is not sufficiently justified, and the absence of important covariates—such as socioeconomic status or home literacy environment—limits the interpretive scope of the findings. Although the dual grouping strategy based on both family risk (FR) and behavioural risk at 60 months (60mDR) adds conceptual depth, its theoretical implications are underdeveloped, particularly in light of divergences between classifications that remain insufficiently explored. The results and discussion sections build coherently on the study’s predictions, confirming associations between ART sensitivity and pre-literacy skills like rapid automatized naming (RAN) and letter knowledge. These findings are consistent with TS theory and suggest ART may serve as an early marker of reading risk. Nevertheless, the interpretation of these associations is at times overstated. The authors frequently imply developmental causality based on correlational data, with limited reporting of statistical details such as effect sizes, model specifications, or confidence intervals. Moreover, findings that barely reach statistical significance (e.g., p = .044, p = .051) are discussed with unwarranted confidence. The discussion rightly acknowledges ART’s specific relation to phonological rather than grammatical skills, supported by the null associations with TROG and CELF measures; however, this specificity is not critically examined in light of alternative explanations, such as shared attentional or cognitive processing mechanisms. While outlier cases are used to illustrate variability in developmental outcomes, they are presented narratively without systematic analysis or integration into the theoretical framework. The discussion would benefit from deeper engagement with competing models, particularly the multiple deficit model, which could offer a more nuanced interpretation of the data than TS theory alone. Critically, the article does not provide a dedicated limitations section, and the few methodological caveats—such as attrition or the limited range of acoustic sensitivity measures—are mentioned only briefly and treated as peripheral rather than central issues. This absence undermines transparency and weakens the reader’s ability to assess the generalizability of the findings. The lack of consideration for environmental or demographic confounds further narrows the explanatory power of the conclusions. Additionally, the prose style, while technically competent, is overly dense and prone to long, complex sentences that may impede comprehension. Overall, the study offers valuable preliminary evidence that ART sensitivity in infancy may relate to later phonological processing and literacy-related outcomes, yet its impact would be significantly enhanced by greater methodological transparency, theoretical pluralism, and a more cautious and critically engaged interpretation of the results.

Author Response

  1. The article presents a compelling longitudinal investigation into the predictive relationship between infant amplitude rise time (ART) sensitivity and pre-reading abilities at 60 months, framed within the Temporal Sampling (TS) theory. While the study is grounded in a strong theoretical framework and addresses a meaningful gap in the literature by linking early auditory processing to later literacy outcomes, several limitations constrain its overall impact.

Author response: We thank the reviewer for this positive assessment of our work. We provide detailed answers to each question and concern raised by the reviewer below.

  1. The introduction, though conceptually thorough, suffers from unnecessary repetition of hypotheses and theoretical expectations, which detracts from clarity and focus.

Author response: Thank you for pointing this out. We have revised the introduction to address this issue. We have removed instances of repetition of our predictions throughout the text, and instead added a final paragraph that succinctly summarises our research questions and hypotheses (as also suggested by Reviewer 3).

  1. Methodologically, the use of established behavioural measures and a rare longitudinal design beginning in infancy are notable strengths. However, the small sample size, particularly for the key longitudinal analyses (N = 24), is not sufficiently justified, and the absence of important covariates—such as socioeconomic status or home literacy environment—limits the interpretive scope of the findings.

Author response: We now include a Limitations section in the Discussion where we provide a more detailed explanation for the limited sample size and discuss options for addressing this issue in future studies. We have also included more information about our participants’ socio-economic status in the Method section.

  1. Although the dual grouping strategy based on both family risk (FR) and behavioural risk at 60 months (60mDR) adds conceptual depth, its theoretical implications are underdeveloped, particularly in light of divergences between classifications that remain insufficiently explored.

Author response: To meet this point, we have added some sentences to prepare the reader and then note the inconsistency of classification during the Conclusion as a limitation. However, as the referee notes elsewhere, we must be careful not to over-interpret the data given these rather small numbers.

  1. The results and discussion sections build coherently on the study’s predictions, confirming associations between ART sensitivity and pre-literacy skills like rapid automatized naming (RAN) and letter knowledge. These findings are consistent with TS theory and suggest ART may serve as an early marker of reading risk. Nevertheless, the interpretation of these associations is at times overstated. The authors frequently imply developmental causality based on correlational data, with limited reporting of statistical details such as effect sizes, model specifications, or confidence intervals. Moreover, findings that barely reach statistical significance (e.g., p = .044, p = .051) are discussed with unwarranted confidence.

Author response: Thank you for raising this concern. We have made several revisions to the discussion to address this point. First, we have re-phrased some interpretations of our findings to avoid overgeneralisations. We kept some discussion that implied developmental causality in our findings given that it was supported by the time-lagged correlations reported in the manuscript. Second, we ensured to report and base our discussion on effect sizes rather than p-values (we report Cohen’s d and Pearson r, which are the relevant effect size measures for the analyses reported in the manuscript). Third, we included a detailed limitations section in the discussion to acknowledge this concern explicitly.  

  1. The discussion rightly acknowledges ART’s specific relation to phonological rather than grammatical skills, supported by the null associations with TROG and CELF measures; however, this specificity is not critically examined in light of alternative explanations, such as shared attentional or cognitive processing mechanisms. While outlier cases are used to illustrate variability in developmental outcomes, they are presented narratively without systematic analysis or integration into the theoretical framework.

Author response: We have added some sentences to this effect regarding cognitive mechanisms; however, we chose a narrative presentation for the outliers as there are only 2 really surprising outliers w.r.t. TS theory.

  1. The discussion would benefit from deeper engagement with competing models, particularly the multiple deficit model, which could offer a more nuanced interpretation of the data than TS theory alone.

Author response: We respectfully disagree with this point. The multiple deficits in the multiple deficit model are all identified once schooling has commenced. The major premise of TS theory is that these multiple deficits can be explained by an earlier-occurring senori-neural atypicality intrinsic to the brain, which affects linguistic processing from infancy onwards. It is the altered linguistic developmental trajectories experienced by affected children that then have multiple cascading effects that lead in later childhood to an apparently diverse range of multiple deficits. By TS theory, these deficits can all be traced back to an auditory-neural cause.

  1. Critically, the article does not provide a dedicated limitations section, and the few methodological caveats—such as attrition or the limited range of acoustic sensitivity measures—are mentioned only briefly and treated as peripheral rather than central issues. This absence undermines transparency and weakens the reader’s ability to assess the generalizability of the findings. The lack of consideration for environmental or demographic confounds further narrows the explanatory power of the conclusions.

Author response: As mentioned above, we have now added a limitations section to the discussion where we describe and justify the main limitations of this study (specifically the limited sample size available for longitudinal analyses) in more detail. We have also expanded the description of our sample, including evidence that our groups were from similar demographic backgrounds and did not differ based on socio-economic status and maternal education.

  1. Additionally, the prose style, while technically competent, is overly dense and prone to long, complex sentences that may impede comprehension.

Author response: Thank you for pointing this out. We have revised the manuscript to simplify complex sentences and improve its flow.

  1. Overall, the study offers valuable preliminary evidence that ART sensitivity in infancy may relate to later phonological processing and literacy-related outcomes, yet its impact would be significantly enhanced by greater methodological transparency, theoretical pluralism, and a more cautious and critically engaged interpretation of the results.

Author response: We hope that the revisions mentioned above address the reviewer’s concerns about our manuscript.

Reviewer 2 Report

Comments and Suggestions for Authors

This is a very interesting and innovative work that proposes a potentially new measure of language development outcomes; early diagnostic of dyslexia and even potentially developmental language disorder. While cross-sectional results for 5-year-olds are very interesting, longitudinal comparison for the amplitude rise time measure was not significant. I do recommend this work for the publication, given the authors address several issues I outline below.

Major:
The main limitation of small sample size for the longitudinal ART –i.e only 24 retaining ART at 10-month scores should be addressed much more clearly in the discussion instead of a short off hand type of sentence on page 13 line 461.
This deserves a separate short paragraph as a gap for future studies with reasons for such high data loss and possible suggestions on how to reduce participant attrition.

Minor:

Materials:

Please provide a short description of the infant ART task instead of making your readers search your previous works. The amplitude rise time discrimination tasks for both infants and 5-year olds are the focus of this paper, so it was rather odd to omit the description of the ART task for infants, even if you described it somewhere else.

Please provide ranges for all the tests in addition to the means and standard deviations. For the letter knowledge you provided the ranges but mean scores and SDs are missing – please add the missing information.

References:

Please check the references numbering I think the count in the paper and in the Reference section is off at least by one starting at 48-50 but please also check earlier parts since I noticed the discrepancy at that point.

Author Response

  1. This is a very interesting and innovative work that proposes a potentially new measure of language development outcomes; early diagnostic of dyslexia and even potentially developmental language disorder. While cross-sectional results for 5-year-olds are very interesting, longitudinal comparison for the amplitude rise time measure was not significant. I do recommend this work for the publication, given the authors address several issues I outline below.

Author response: We thank the reviewer for these positive comments. We provide detailed answers to each point raised by the reviewer below.

  1. The main limitation of small sample size for the longitudinal ART –i.e only 24 retaining ART at 10-month scores should be addressed much more clearly in the discussion instead of a short off hand type of sentence on page 13 line 461. This deserves a separate short paragraph as a gap for future studies with reasons for such high data loss and possible suggestions on how to reduce participant attrition.

Author response: We now include a limitations section in the discussion where we provide a more detailed explanation for the limited sample size and discuss options for addressing this issue in future studies.

  1. Materials: Please provide a short description of the infant ART task instead of making your readers search your previous works. The amplitude rise time discrimination tasks for both infants and 5-year olds are the focus of this paper, so it was rather odd to omit the description of the ART task for infants, even if you described it somewhere else.

Author response: We have now included this information in the manuscript.

  1. Please provide ranges for all the tests in addition to the means and standard deviations. For the letter knowledge you provided the ranges but mean scores and SDs are missing – please add the missing information.

Author response: We have now included this information in Tables 1 and 3.

  1. References: Please check the references numbering I think the count in the paper and in the Reference section is off at least by one starting at 48-50 but please also check earlier parts since I noticed the discrepancy at that point.

Author response: Thank you for noticing this error. We have now corrected the references numbering throughout the manuscript.

Reviewer 3 Report

Comments and Suggestions for Authors

Please check in-text comments of the reviewer, add literature review section, address the comments regarding the study, methodology, elaborate on each test and procedure of data collection and data analysis, please add explicit research questions and address them in the discussion section.

Author Response

  1. Please check in-text comments of the reviewer, add literature review section, address the comments regarding the study, methodology, elaborate on each test and procedure of data collection and data analysis, please add explicit research questions and address them in the discussion section.

Author response: Below we transcribe the comments provided by the reviewer in the manuscript and provide a response to each comment.

  1. [Introduction] Please add the structure of the article; please add lit review section; please add explicitly stated research questions.

Author response: We have revised the structure of the introduction section to address these suggestions by the reviewer. We have not created a section specifically called “Literature Review” since this information is included in the “Introduction” section. However, we did create “The Present Study” section, which now clearly outlines the research questions and hypotheses for this study, which set the structure for the rest of the article.

  1. [Participants] Please add information about the country where the data were collected, in which settings; please explain the sampling procedure--how the researchers got access to the participants contacted parents etc?

Author response: This information is now included in the manuscript.

  1. [Methods] Materials and procedure--add this section; data collection, data analysis

Author response: We thank the reviewer for this suggestion. However, we have not included these separate section titles since they do not fit the structure of our manuscript. We explain the materials, apparatus, procedure for data collection and data analysis for each type of task described in the Method.

  1. [Infant rise time task] Still here some description of the task is needed.

Author response: A more detailed description of the task is now provided.

  1. what is the SES of the families, the parents of the children? please provide this information.

Author response: This information is now included in the Participants section.

  1. Please provide more information about the test, number of sentences, linguistic phenomena (Sentence Repetition); please provide more description, number of items and grammatical phenomena measured (TROG); what these 13 items includes, please provide this information (Letter Knowledge); it would be nice to have a table with all the test and measures and phenomena tested (re. Standardised tests); how many items, what they measure (receptive vocabulary); please provide more information about the number of items and measures (composite vocabulary); please provide more information about this test (non-verbal IQ)

Author response: We thank the Reviewer for these suggestions. All these sections describe standardised tests, and it is not conventional to provide a detailed description of the specific items of each test. We provide relevant references to the test manuals and accompanying articles, which include detailed descriptions of the test items, scoring procedures, and norming samples for interested readers.   

  1.        
    [Discussion] please add research questions to the study section and then address these research questions in the discussion section; more elaboration is needed, several RQs or sub-questions; you need to have several research questions and address them here in the discussion section and use sections and sub-sections.

Author response: We have revised the first paragraph of the discussion to provide a clearer outline of our research questions and initial hypotheses. While we have chosen not to create further sub-sections in the Discussion, we have ensured that each paragraph starts by clearly stating the findings that it will discuss, thus providing a roadmap for our readers.

  1. please elaborate more on the conclusion

Author response: Overall, the discussion section is now more succinct as we incorporate more points about limitations and individual differences.

Round 2

Reviewer 1 Report

Comments and Suggestions for Authors

Thank you for your efforts!